# Whole Blood Spermine/Spermidine Ratio as a New Indicator of Sarcopenia Status in Older Adults

**DOI:** 10.3390/biomedicines11051403

**Published:** 2023-05-09

**Authors:** Hidenori Sanayama, Kiyonori Ito, Susumu Ookawara, Takeshi Uemura, Yoshio Sakiyama, Hitoshi Sugawara, Kaoru Tabei, Kazuei Igarashi, Kuniyasu Soda

**Affiliations:** 1Division of Neurology, First Department of Integrated Medicine, Saitama Medical Center, Jichi Medical University, Saitama 330-8503, Japan; 2Division of Nephrology, First Department of Integrated Medicine, Saitama Medical Center, Jichi Medical University, Saitama 330-8503, Japan; 3Department of Pharmaceutical Sciences, Faculty of Pharmaceutical Sciences, Josai University, Saitama 330-0295, Japan; 4Division of General Medicine, Department of Comprehensive Medicine 1, Saitama Medical Center, Jichi Medical University, Saitama 330-8503, Japan; 5Department of Internal Medicine, Minamiuonuma City Hospital, Niigata 949-6680, Japan; 6Amine Pharma Research Institute, Innovation Plaza at Chiba University, Chiba 260-0856, Japan; 7Saitama Medical Center, Jichi Medical University, Saitama 330-8503, Japan; 8Saitama Ken-o Hospital, Saitama 363-0008, Japan

**Keywords:** polyamine, spermine, spermidine, sarcopenia, skeletal muscle index, biomarker

## Abstract

Early diagnosis and therapeutic intervention improve the quality of life and prognosis of patients with sarcopenia. The natural polyamines spermine and spermidine are involved in many physiological activities. Therefore, we investigated blood polyamine levels as a potential biomarker for sarcopenia. Subjects were Japanese patients >70 years of age who visited outpatient clinics or resided in nursing homes. Sarcopenia was determined based on muscle mass, muscle strength, and physical performance according to the criteria of the Asian Working Group for Sarcopenia (2019). The analysis included 182 patients (male: 38%, age: 83 [76–90] years). Spermidine levels were higher (*p* = 0.002) and the spermine/spermidine ratio was lower (*p* < 0.001) in the sarcopenia group than in the non-sarcopenia group. Polyamine concentration analysis showed that the odds ratios for age and spermidine changed in parallel with sarcopenia progression, and the odds ratio for the spermine/spermidine ratio changed inversely with the degree of sarcopenia progression. Additionally, when the odds ratio was analyzed with spermine/spermidine instead of polyamine concentrations, only for spermine/spermidine, the odds ratio values varied in parallel with the progression of sarcopenia. Based on the present data, we believe that the blood spermine/spermidine ratio may be a diagnostic indicator of risk for sarcopenia.

## 1. Introduction

Sarcopenia, defined as the age-related loss of muscle mass, strength, and physical performance, is a progressive, generalized musculoskeletal disorder that is associated with an increased risk of adverse events, such as falls, fractures, and impaired mobility [1]. It is also associated with cardiac, respiratory, neurodegenerative, and cognitive complications that increase the need for nursing home care and reduce life expectancy [2,3,4,5]. As such, sarcopenia reflects senescence, a typical pathology associated with aging. Therefore, prevention, early diagnosis, and early therapeutic intervention will improve the quality of life and prognosis of elderly patients. However, diagnosing sarcopenia can be difficult because (1) multiple factors must be assessed, including muscle mass, strength, gait, and other physical functions, and (2) different reference values exist for men and women. Many older people will need care in the future, and older people with limited mobility are more likely to experience early onset frailty than healthy adults [6]. Therefore, it is important to have a simple and accurate method for the prediction of sarcopenia risk.

In recent years, the relationship between blood polyamines and various health conditions has been noted [7,8,9,10,11]. The natural polyamines spermine (SPM) and spermidine (SPD) and their precursor putrescine (PUT) are low molecular weight aliphatic amines that contain multiple amino groups. Polyamines, which are found in all organisms including humans, are essential for cell growth and differentiation and are involved in many physiological activities [12]. Intracellular polyamine concentrations are altered by the activity of intracellular polyamine synthases and catabolites, as well as by polyamine supply from outside the cell and polyamine efflux from the cell. Enzymatic activities for polyamine synthesis decrease with age [12], suggesting that polyamine levels decrease with age. In fact, when all age groups, including children, are examined, blood polyamine levels decrease with age [13]. However, the age-related decline in tissue polyamine concentrations is observed only in early life (fetal and developmental periods) [14]. No significant decrease in blood polyamine concentrations was observed in adults. However, several reports in humans have shown that the blood SPM concentration relative to blood SPD (SPM/SPD ratio) tends to decrease because SPM concentrations show a decreasing trend and SPD shows an increasing trend [15,16,17,18].

Blood polyamine levels are influenced by the supply of polyamines from outside the cells. For example, we have previously reported that a polyamine-rich diet increases blood levels of SPM, which has much stronger biological activities than SPD. This increase was accompanied by evidence of SPM bioactivity, such as suppression of age-related pro-inflammatory conditions in mice and humans [15,19]. In addition, we have shown that a long-term increase in polyamine intake prolongs the life span of mice [20]. We also found that SPM activates DNA methyltransferases and inhibits the progression of DNA methylation abnormalities associated with polyamine deficiency and aging [21,22]. The increase in blood SPM concentration and the concomitant increase in the SPM/SPD ratio were associated with a positive health status, which subsequently contributed to an increased healthy life expectancy.

Conversely, blood levels of SPD are elevated in patients with several age-related conditions, including impaired cognitive function [7,9] and neurodegenerative diseases such as Alzheimer’s disease [10] and Parkinson’s disease [11]. In fact, the age-related decline in the SPM/SPD ratio is accelerated in these patients [8]. The mechanism of the increase in SPD levels and the decrease in SPM/SPD ratio is not well understood; however, it is necessary to consider that an increase in the SPD levels is a complementary mechanism that improves the disease state by stimulating autophagy activation in the brain. One reason for this is that the basic properties of polyamines were not understood in the experiments investigating the activation of autophagy by SPD, and thus artifacts other than the bioactivity of SPD itself were considered bioactive [23]. In addition, the blood–brain barrier prevents polyamines from entering brain tissue [12]. Therefore, we speculate that increased SPD concentrations and a decreased SPM/SPD ratio may lead to the development and progression of lifestyle-related diseases that worsen with age.

Sarcopenia is a typical condition that develops and progresses with age. While previous research has shown that blood polyamines are associated with age-related diseases, no previous study has examined whether they are associated with the typical age-related pathological condition, sarcopenia. In addition, a simple diagnostic approach to predict the risk of developing sarcopenia has not yet been developed. Therefore, in this study, we investigated whether blood polyamine levels are associated with sarcopenia and whether blood polyamine levels can be used as a biomarker for the early detection or assessment of progression of sarcopenia.

## 2. Materials and Methods

### 2.1. Study Design and Participants

In this cross-sectional study, we included elderly persons aged 70 years or older who visited outpatient clinics affiliated with Minamiuonuma City Hospital (Minamiuonuma City, Japan) or who resided in nursing homes for the elderly (Minami-en and Maiko-en, Minamiuonuma, Japan). A flowchart of the patient selection process is shown in Figure 1. Patients with pacemakers were excluded because bioelectrical impedance analysis (BIA) cannot be used to measure muscle mass in this population. Since neoplasms have a significant impact on polyamine levels, patients with cancer and those with a history of cancer treatment within 3 months were excluded.

### 2.2. Data and Sample Collection

Medical history and clinical information such as age, history of cardiovascular disease (CVD) and cerebrovascular disease (CeVD), history of dementia and fractures were obtained from medical records, physical examination results, and questionnaires. Height and weight were measured in the standing position. However, the height of participants who were unable to stand was measured with a tape measure. Measurements were taken from the top of the head to the heel. The weight of the participants who were unable to stand was calculated by subtracting the weight of the wheelchair from the value measured in the wheelchair. Body mass index (BMI) was calculated as weight (kilograms) divided by height (meters squared). For the measurement of body composition and muscle mass, a multi-frequency BIA was performed using the InBody S10 (InBody Japan, Tokyo, Japan). The skeletal muscle mass index (SMI) was calculated by dividing the skeletal muscle mass of the limbs in kilograms by the square of the height in meters. Hand grip strength (HGS) was measured with a hand grip strength meter (Smedley’s Hand Dynamometer; Matsumiya Ika Seiki, Tokyo, Japan) while the patient was seated, and the higher value of the left- and right-hand grip strength tests was used. The walking speed (WS) was calculated as the time required to walk 10 m at a self-selected WS.

### 2.3. Definition of Sarcopenia

We defined sarcopenia as a decrease in muscle mass and a decrease in muscle strength or physical performance according to the diagnostic criteria established by the Asian Working Group for Sarcopenia (2019) [1]. Thus, a diagnosis of sarcopenia was made based on the following: SMI < 7.0 kg/m^2^ and <5.7 kg/m^2^ for men and women, respectively, plus either an HGS value of <28 kg and <18 kg for men and women, respectively, or WS less than 1.0 m/s. Participants who did not meet the criteria for sarcopenia were assigned to the non-sarcopenia group. For further analysis, the non-sarcopenia group was subdivided as follows: a semi-sarcopenia group (patients who had either a decrease in SMI or a decrease in HGS and/or WS but who did not meet the criteria for a sarcopenia diagnosis) and a healthy group (the remaining patients from the non-sarcopenia group).

### 2.4. Blood and Biochemical Tests

Blood samples were collected from participants at room temperature. Whole blood samples for measurement of polyamine concentrations were immediately stored at −20 °C until assayed. All measurements of blood samples, except for polyamines, were performed in a clinical laboratory at Minamiuonuma City Hospital. Hemoglobin concentration (g/dL) was measured via the sodium lauryl sulfate hemoglobin detection method using an automated blood cell analyzer (XN-2000; Sysmex Corporation, Kobe, Japan). An automated biochemical analyzer (BM6070; JEOL Ltd., Akishima, Japan) was used to perform biochemical blood tests. The concentration of albumin (Alb) in serum (g/dL) was measured using the modified bromocresol purple method. Creatinine levels were measured using the enzymatic method, and the estimated glomerular filtration rate (eGFR; mL/min/1.73 m^2^) was calculated using the following formulas: eGFR (mL/min/1.73 m^2^) = 194 × creatinine ^−1.094^ × age ^−0.287^ (for men); eGFR (mL/min/1.73 m^2^) = 194 × creatinine ^−1.094^ × age ^−0.287^ × 0.739 (for women). Low-density lipoprotein cholesterol (LDL-C) (mg/dL) and high-density lipoprotein cholesterol (mg/dL) levels were measured via direct assay. Triglycerides (TG) (mg/dL) were measured using the enzymatic method. Hemoglobin A1c (HbA1c) (%) was measured using high-performance liquid chromatography (HPLC) (AH8290; ARCRAY Inc., Kyoto, Japan).

### 2.5. Determination of Polyamine Concentrations in Whole Blood

Whole blood samples were heparinized, collected, and stored at −20 °C or below. Whole blood was thawed and degraded using sonication and freeze–thaw cycles to measure polyamine concentrations. Polyamine concentrations were determined via HPLC [24] at the Cardiovascular Institute for Medical Research, Saitama Medical Center, Jichi Medical University. For polyamine extraction, whole blood was diluted fivefold with 5% trichloroacetic acid (TCA) and incubated at 95 °C for 45 min. After centrifugation at 13,000× *g* for 20 min at 4 °C, the supernatant was collected and deproteinized by increasing the TCA concentration to 10% and incubating at 95 °C for 45 min, followed by centrifugation at 13,000× *g* for 20 min at 4 °C. Polyamines in 20 µL of the TCA supernatant were separated with an HPLC system (Shimadzu Corporation, Kyoto, Japan) using a TSKgel Polyaminepak column (column size 4.6 mm ID × 50 mm length, particle size 7 µm, TOSOH Bioscience, Tokyo, Japan) at 50 °C. The flow rate was set at 0.42 mL/min and the composition of the separation buffer adjusted to pH 5.10 was 0.09 M citric acid (NACALAI TESQUE, INC., Kyoto, Japan), 2 M NaCl (NACALAI TESQUE, INC., Kyoto, Japan), 0.64 mM n-capric acid (NACALAI TESQUE, INC., Kyoto, Japan), 0.1% Brij-35 (Sigma-Aldrich Japan, Tokyo, Japan), 20% methanol (FUJIFILM Corporation, Osaka, Japan). Polyamines were detected via fluorescence intensity after the column effluent was reacted with the solution containing 0.4 M boric buffer (pH 10.4) (NACALAI TESQUE, INC., Kyoto, Japan), 0.1% Brij-35, 2.0 mL/L 2-mercaptoethanol (NACALAI TESQUE, INC., Kyoto, Japan), and 0.06% o-phthalaldehyde (NACALAI TESQUE, INC., Kyoto, Japan) at 50 °C. The flow rate of o-phthalaldehyde solution was 0.42 mL/min and fluorescence was measured at an excitation wavelength of 340 nm and an emission wavelength of 455 nm. The retention time was 12 min for SPD and 23 min for SPM. Concentrations in the original whole blood samples are expressed in micromoles.

Blood SPD, SPM, and SPM/SPD ratio were each measured three consecutive times in two samples and the average was calculated as the %CV (coefficient of variation). The %CVs were 4.3% for SPD, 3.6% for SPM, and 6.7% for the SPM/SPD ratio.

### 2.6. Statistical Analysis

The Shapiro–Wilk test was used to determine whether the variables were normally distributed. Data are presented as mean ± standard deviation for normally distributed data, and median and interquartile range for non-normally distributed data. To evaluate differences in variables between the sarcopenia and non-sarcopenia groups, normally distributed variables were evaluated using the unpaired Student’s *t*-test, and non-normally distributed variables were evaluated using the Mann–Whitney U test.

Comparisons between the three groups were made as follows. The z-test was used for multiple comparisons of proportions, and the Bonferroni method was used to determine significance. Specifically, the Bonferroni method was used to obtain the “adjusted significance level (*p*′)”, and a judgment was made at *p*′ = 0.017 for the probability value of the significance test (z-test) results for each pair of comparisons. Results of the comparison between groups were considered significantly different at *p* < 0.05. The Kruskal–Wallis test was used for multiple comparisons of values. Levene’s test was used to assess homogeneity of variance. Tukey’s method was used to determine between-group differences for variables with equal variance, and Games-Howell’s method was used for variables with unequal variance.

Correlations between age and SPD, SPM, or the SPM/SPD ratio were evaluated by using Spearman’s rank correlation coefficient. Multivariate logistic regression models were used to calculate adjusted odds ratios (ORs) with 95% confidence intervals (CIs) for the risk of sarcopenia associated with the SPD and SPM concentrations or the SPM/SPD ratio. Because SMI, HGS, and WS are diagnostic criteria for sarcopenia and nursing home occupancy rates reflect the consequences of developing of sarcopenia, analyses were performed after excluding these factors.

All analyses were performed using IBM SPSS Statistics for Windows version 28.0 (IBM, Armonk, NY, USA). A two-tailed *p*-value < 0.05 was considered statistically significant.

## 3. Results

### 3.1. Participant Characteristics

A total of 270 patients were included; 9 were excluded because they either died or were transferred to another hospital before the study began. We could not accurately measure the muscle mass of 53 patients because they were unable to maintain the desired position. In addition, after excluding 26 patients with a confirmed history of cancer, 182 patients were included in the analysis. One-hundred and nineteen patients were hospital outpatients, and sixty-three were nursing home residents (Figure 1).

### 3.2. Background of the Study Participants

Participant characteristics are summarized in Table 1. Of the 182 participants (68 men and 114 women), 111 did not meet the diagnostic criteria for sarcopenia (non-sarcopenia group; 52 men and 59 women). The remaining 71 comprised the sarcopenia group (16 men and 55 women), which had a higher proportion of women (77% vs. 53%, *p* < 0.001) than the non-sarcopenia group. In addition, participants in the sarcopenia group were older than those in the non-sarcopenia group (*p* < 0.001). The percentage of nursing home admissions was higher in the sarcopenia group than that in the non-sarcopenia group (80% vs. 5%, *p <* 0.001). Fifty-six (89%) of the nursing home participants had difficulty walking. Participants in the sarcopenia group had a higher incidence of CeVDs, dementia, and fractures than those in the non-sarcopenia group; however, there was no significant difference in CVD history rates. SMI, WS, and HGS, which are the diagnostic elements of sarcopenia and its associated indices, as well as BMI, were all significantly lower (*p* < 0.001) in the sarcopenia group than in the non-sarcopenia group.

The results showed higher SPD concentrations (*p* = 0.002) in the sarcopenia group than in the non-sarcopenia group, but SPM concentrations were not significantly different (*p* = 0.701), and the SPM/SPD ratio was lower (*p* < 0.001) in the sarcopenia group. Hemoglobin, Alb, TG, and HbA1c were all lower in the sarcopenia group than in the non-sarcopenia group (*p* < 0.001), but eGFR was higher in the sarcopenia group (*p* = 0.046).

### 3.3. Blood Polyamines

SPD and SPM concentrations and the SPM/SPD ratios were compared between the sarcopenia and non-sarcopenia groups (Figure 2). The SPD concentration was significantly higher in the sarcopenia group than in the non-sarcopenia group (*p* = 0.002), and the SPM/SPD ratio was significantly lower in the sarcopenia group than in the non-sarcopenia group (*p* < 0.001). In addition, SPM concentrations were not significantly different between the sarcopenia and non-sarcopenia groups (*p* = 0.701).

Figure 3 shows the correlations between age and polyamine concentrations and SPM/SPD ratios in the sarcopenia and non-sarcopenia groups. A non-significant association (*p* = 0.070) between increased SPD levels and age was observed in the sarcopenia group, but not in the non-sarcopenia group (*p* = 0.339). A trend of increasing SPD levels with age was observed in the sarcopenia group (ρ= 0.216, *p* = 0.070), but not in the non-sarcopenia group (ρ = −0.092, *p* = 0.339). SPM concentrations showed no relationship with age in both groups and remained almost unchanged (non-sarcopenia group (ρ = −0.003, *p* = 0.979), and sarcopenia group (ρ = 0.030, *p* = 0.802)). In the sarcopenia group, the SPM/SPD ratio had a non-significant negative correlation value with age (ρ = −0.194, *p* = 0.106). However, in the non-sarcopenia group, no such trend in age-related changes was observed in the SPM/SPD ratio (ρ = 0.065, *p* = 0.495).

In the sarcopenia group, there was a positive correlation between age and the SPD concentration, and a negative correlation between age and the SPM/SPD ratio; however, both correlations were not statistically significant. Spearman’s rank correlation coefficient (ρ) was used. SPD—spermidine; SPM—spermine.

### 3.4. Binomial Multivariate Logistic Regression Analysis

Binomial multivariate logistic regression analysis was performed to select parameters associated with sarcopenia progression. The non-sarcopenia group was used as the reference for the analysis, and ORs (95% confidence intervals) were calculated for each explanatory variable. ORs were calculated using all explanatory variables listed in Table 2 and Table 3. Table 2 shows the ORs associated with sarcopenia for each variable, including SPD and SPM concentrations. SPD concentration was associated with increased risk of sarcopenia (OR = 1.481 [95% CI: 1.073–2.044]), and SPM concentration was associated with decreased risk of sarcopenia (OR = 0.502 [95% CI: 0.299–0.842]). Other factors associated with increased risk of sarcopenia included age, a history of dementia, and eGFR, and factors associated with decreased risk of sarcopenia included BMI, Alb, and HbA1c. Table 3 shows the ORs associated with sarcopenia for each variable, including SPM/SPD ratio instead of SPM and SPD concentration in Table 2. The SPM/SPD ratio was associated with a reduced risk of sarcopenia (OR = 0.033 [95% CI: 0.002–0.557]). Other factors associated with increased risk of sarcopenia included age and eGFR, and factors associated with decreased risk of sarcopenia included BMI, Alb, and HbA1c.

### 3.5. Factors Associated with Sarcopenia Progression

Data for the healthy, semi-sarcopenia, and sarcopenia groups are shown in Table 4. Compared to the healthy group, there were significantly more women in the semi-sarcopenia and sarcopenia groups (*p* < 0.05). As sarcopenia progressed, i.e., from normal to semi-sarcopenia, and then to sarcopenia, age increased (*p* < 0.001), nursing home placement and the preexisting CeVD and dementia were more common (*p* < 0.05), and SMI, HGS, WS, and Alb were lower (*p* < 0.001). BMI was not significantly different between the healthy and semi-sarcopenia groups (*p* = 0.915); however, the differences were significant between the healthy and sarcopenia groups and between the semi-sarcopenia and sarcopenia groups (*p* < 0.001). SPD concentration was significantly higher in the sarcopenia group than in the healthy group (*p* < 0.001), but there were no significant differences between the healthy and semi-sarcopenia groups (*p* = 0.087) or between the semi-sarcopenia and sarcopenia groups (*p* = 0.087). The SPM/SPD ratio was significantly lower in the sarcopenia group than in the healthy group (*p* = 0.010), but not significantly different between the healthy and semi-sarcopenia groups (*p* = 0.772) and between the semi-sarcopenia and sarcopenia groups (*p* = 0.087).

Because SMI, HGS, and WS are diagnostic criteria for sarcopenia and nursing home occupancy rates are a consequence of sarcopenia, multinomial multivariate logistic regression analyses were performed with respect to the healthy group after excluding these factors. Table 5 shows the ORs associated with semi-sarcopenia or sarcopenia relative to the healthy group for each variable, including SPD and SPM concentrations. Due to the loss of HbA1c data in one case, 54 cases were analyzed for semi-sarcopenia. Whole blood SPD concentrations increased with the severity of sarcopenia; OR = 1.625 [95% CI: 1.110–2.379] for semi-sarcopenia and OR = 2.218 [95% CI: 1.393–3.533] for sarcopenia. Several other factors were also associated with an increased or decreased risk of semi-sarcopenia (sex [M/F ratio], Age, CeVD prevalence) or sarcopenia (Alb, Age, eGFR, LDL-C, HbA1c). However, the only significantly different variables associated with the different severities of sarcopenia were age and SPD concentration. For both variables, the OR was increased in the sarcopenia group compared to the semi-sarcopenia group.

Table 6 shows the results of the multinomial multivariate logistic regression analysis results using the SPM/SPD ratio instead of SPM and SPD concentrations. As shown in Table 6, due to the loss of HbA1c data in one case, 54 cases were analyzed in semi-sarcopenia. The SPM/SPD ratio was associated with a reduced risk of semi-sarcopenia (OR = 0.057 [95% CI: 0.004–0.796]) and sarcopenia (OR = 0.002 [95% CI: 0.001–0.091]). Several other factors were also associated with an increased or decreased risk of semi-sarcopenia (sex [M/F ratio], age, CeVDs, TG) and sarcopenia (age, CeVDs, dementia, Alb, HbA1c). Age had a higher OR for risk of sarcopenia compared to semi-sarcopenia. On the other hand, the SPM/SPD ratio had a lower OR for risk of sarcopenia compared to semi-sarcopenia. Thus, aging and a decrease in the SPM/SPD ratio were found to be associated with the progression of sarcopenia.

## 4. Discussion

Previous studies of age-related changes in blood polyamine concentrations in individuals under the age of 80 have shown a decreasing trend in SPM/SPD, although the difference is not statistically significant [15,16,17,18]. In this study, we observed a tendency for the SPM/SPD ratio to decrease with age in the sarcopenia group; this tendency was not observed in the non-sarcopenia group. Additionally, the data showed that the SPD concentration was significantly higher and the SPM/SPD ratio was significantly lower in patients with sarcopenia than in those without sarcopenia. Elevated blood levels of SPD have been reported in patients with impaired cognitive function [7,9] and several neurodegenerative diseases [10,11]. In addition, a decreased SPM/SPD ratio has been reported in neurodegenerative diseases such as Alzheimer’s and Parkinson’s disease [8].

Blood polyamine levels reflect systemic polyamine concentrations. SPD has been found to be present at higher concentrations than SPM in many mammalian organs and tissues, as reflected by the low SPM/SPD ratio in mouse tissues [25]. However, in the same study, the concentration of SPM was comparable to that of SPD in several murine organs. In addition, SPM/SPD ratios were higher in the brain and muscle than in other organs. Although the blood–brain barrier is thought to block the entry of polyamines from the blood into the brain, changes in polyamine concentrations in the brain are thought to be reflected in blood polyamine concentrations via the spinal fluid. Sarcopenia has been associated with decreased muscle mass and brain volume in humans [26]. This implies cell disintegration, and the polyamines released from within the cell as a result would be reflected in the blood levels. Although the SPM/SPD ratio should increase as polyamines leak from brain and muscle cells where SPM is abundant, our data show progressively lower SPM/SPD ratios in the following order: healthy, semi-sarcopenia, and sarcopenia. This finding suggests that changes in polyamine levels in patients with sarcopenia are not simply caused by cell destruction.

Inflammation, which is involved in the onset and progression of senescence and age-related pathologies, activates enzymes such as spermidine/spermine *N*^1^-acetyltransferase (SSAT) and spermine oxidase (SMO) [27,28]. SSAT, together with acetylpolyamine oxidase (AcPAO), converts SPM to SPD and simultaneously converts SPD to PUT. On the other hand, SMO primarily converts SPM to SPD but does not act on SPD (Figure 4). Overall, the induction of inflammation causes more SPM to be degraded. In contrast with non-inflammatory conditions, inflammatory conditions result in significantly greater SPM degradation. This may be the mechanism responsible for the increased SPD concentration and decreased SPM/SPD ratio observed in the sarcopenia group. It is also important to note that neurodegenerative diseases that have been linked to polyamines have also been found to be closely associated with chronic inflammation [29,30,31]. Additionally, patients with these diseases have been found to have elevated blood SPD levels and decreased SPM/SPD ratios [7,9,10,11].

Again, the increase in SPD was unlikely to be due to a compensatory increase in absorption from the gastrointestinal tract, as there were no specific dietary or other interventions for patients with sarcopenia. Even if we assume that elevated SPD exerts a feedback function to promote physiological activity in patients with sarcopenia, there is no scientific evidence to explain this. First, there is little evidence of reduced SPD in tissues and organs of neurodegenerative diseases and similar age-related diseases. Second, it has been shown that a diet high in spermidine does not lead to an increase in spermidine concentrations [7,15,21]. Third, because the blood–brain barrier prevents polyamines in the blood from entering the brain. Even if blood concentrations increase by a feedback mechanism due to a decrease in brain concentrations, polyamines outside the brain will not affect polyamine concentrations in the brain and will not exert any biological activity.

AcPAO oxidizes acetylated polyamines converted through SSAT, producing H_2_O_2_ and 3-acetoamidopropanal (3-AAP) as by-products. SMO directly converts SPM to SPD to produce H_2_O_2_ and the aldehyde 3-aminopropanal (3-AP) (Figure 4). The produced 3-AP is spontaneously deaminated to produce acrolein, a highly toxic aldehyde [32]; however, little acrolein is produced from 3-AAP [33]. It has been reported that acrolein and 3-AP are highly cytotoxic, whereas 3-AAP is not [34]. Findings suggesting inflammation-induced activation of SMO to degrade SPM and produce acrolein include increased acrolein levels in renal failure [35], cerebrovascular disease [36], and other age-related conditions that have been reported to have adverse health effects [37]. In addition to inflammation-induced oxidative stress, the cytotoxic activities of 3-AP and acrolein produced by SMO may play an important role in the progression of sarcopenia and many other age-related diseases.

There are several reasons why we measured polyamine concentrations in whole blood rather than in serum or plasma. First, it can be very difficult to accurately detect polyamines, especially SPM levels, in the serum or plasma. This is because most polyamines in the blood are found in blood cells [38]. Accurate measurement of SPM concentration at low levels can be very difficult because HPLC cannot distinguish between SPM peaks and noise that occurs as baseline variation [39]. The second is to eliminate the effect of hemolysis on serum and plasma polyamine levels. When hemolysis occurs in a blood sample, it releases polyamines, which are present in large quantities in blood cells, and even a very small amount can have a significant effect on polyamine concentrations. Measuring polyamine levels in whole blood requires special techniques, but we have established a method to ensure accurate detection.

Due to the large inter-individual variability in polyamine concentrations [12], it may be difficult to identify a cut-off value at which sarcopenia risk can be determined with a single measurement. Additionally, because various factors related to the conditions under which polyamine concentrations are measured can cause differences in measured polyamine concentrations, we believe it is difficult to compare and evaluate values measured at different times and to track progress. However, the SPM/SPD ratio can be measured as a relative value with a single measurement. Therefore, the data are reliable for individual and long-term comparisons, even if absolute polyamine concentrations vary due to differences in measurement methods. As a result, we believe that observing the time-dependent changes in the SPM/SPD ratio, which decreases as sarcopenia progresses, can be used to determine the risk of developing sarcopenia.

We found that the SPD concentration was significantly higher and the SPM/SPD ratio was significantly lower in patients with sarcopenia than in those without sarcopenia, and these levels were also found to change with the severity of sarcopenia. Due to the large individual variability in polyamine levels, it is difficult to identify a cut-off value at which the risk of developing sarcopenia can be determined. However, in conclusion, we believe that the risk of developing sarcopenia can be determined by observing changes in the blood SPM/SPD ratio over time, which is a relative value.

## 5. Conclusions

The study first showed the inverse relationship between blood spermine concentration relative to spermidine concentration (SPM/SPD ratio) and the degree of sarcopenia. Additionally, we also discussed the possible pathological background. Since several reports showed the inverse relationship between blood SPM/SPD ratio and the degree of age-related diseases, blood SPM/SPD ratio may be an indicator to predict or detect age-related pathological changes.

## Figures and Tables

**Figure 1 biomedicines-11-01403-f001:**
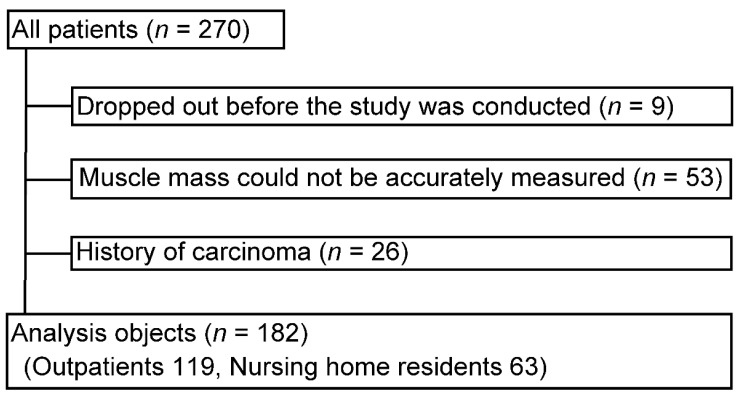
Flow diagram of the patient selection process.

**Figure 2 biomedicines-11-01403-f002:**
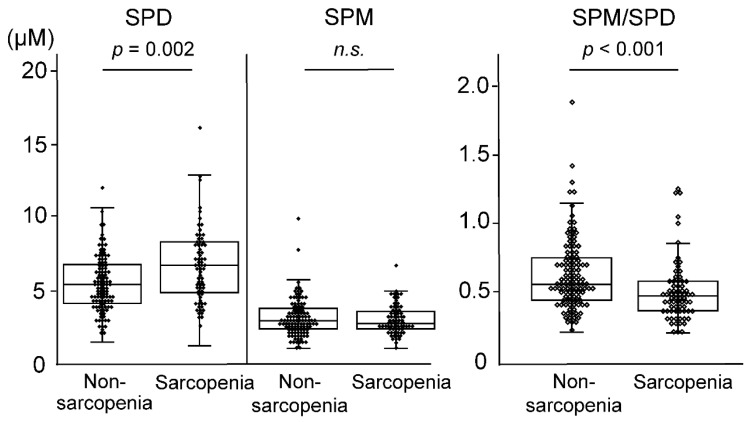
Comparison of blood polyamine concentrations in participants with and without sarcopenia. Box and whisker plots and scatter plots of the whole-blood SPD concentration, SPM concentration, and SPM/SPD ratio in the non-sarcopenia and sarcopenia groups. For each box, the interior line shows the median, and the edges of the box are estimates of the first and third quartiles. The whiskers extend to the most extreme data points that are not considered outliers. The black dots indicate the value for each participant. Whole-blood polyamine concentrations were measured by high-performance liquid chromatography. SPD, spermidine; SPM, spermine.

**Figure 3 biomedicines-11-01403-f003:**
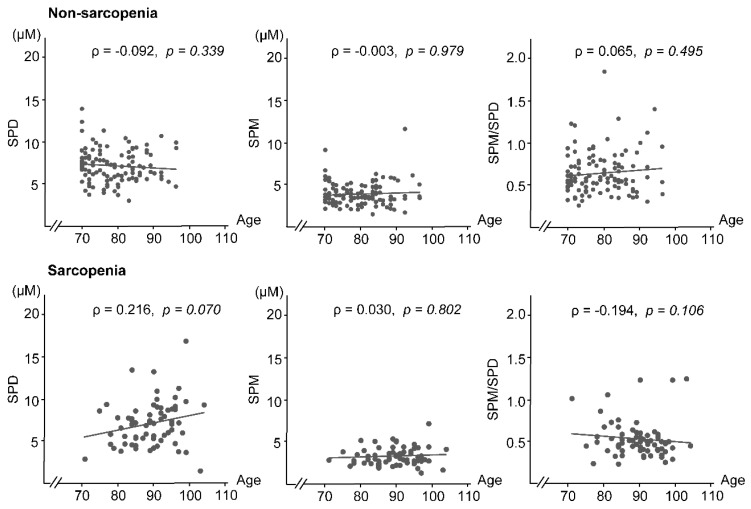
Aging-associated changes in SPD and SPM concentrations and the SPM/SPD ratio in the non-sarcopenia (**upper** figures) and sarcopenia (**lower** figures) groups.

**Figure 4 biomedicines-11-01403-f004:**
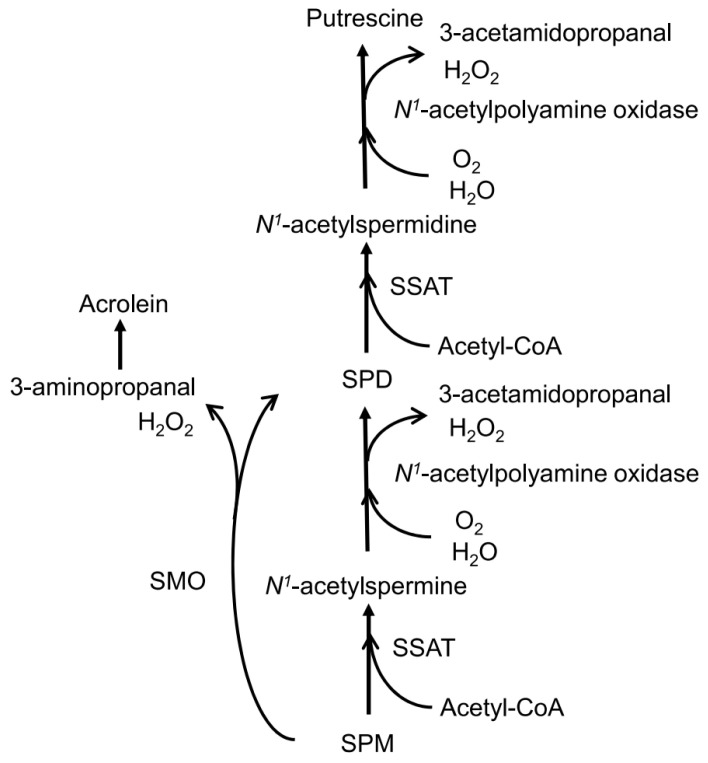
Inflammation-induced enzyme activation for polyamine catabolism and the production of byproducts. Substances that provoke inflammation also activate SSAT and SMO. SMO does not act on spermidine but rather degrades spermine to produce substances with strong cytotoxic activity, such as acrolein. SMO—spermine oxidase; Acetyl-CoA—acetyl coenzyme A; SSAT—spermidine/spermine-*N*^1^-acetyltransferase; SPM—spermine; SPD—spermidine.

**Table 1 biomedicines-11-01403-t001:** Characteristics of the study participants.

**Characteristics**	**All Patients**	**Non-Sarcopenia**	**Sarcopenia**	** *p* ** **Value ***
No. of participants (M/F)	182 (68/114)	111 (52/59)	71 (16/55)	<0.001
Age (years)	83 (76–90)	78 (73–84)	90 (84–95)	<0.001
No. of N.H. residents (%)	63 (35)	6 (5)	57 (80)	<0.001
Past history (%)				
CVDs	8 (4)	6 (5)	2 (3)	0.407
CeVDs	39 (21)	12 (11)	27 (38)	<0.001
Dementia	71 (39)	16 (14)	55 (77)	<0.001
Bone fracture	23 (13)	8 (7)	15 (21)	0.006
BMI (kg/m^2^)	22.2 ± 3.5	23.8 ± 2.7	19.7 ± 3.1	<0.001
SMI (kg/m^2^)	6.5 (4.9–8.0)	7.7 (6.5–8.5)	4.6 (3.6–5.3)	<0.001
WS (m/s)	0.87 (0–1.16)	1.1 (0.87–1.28)	0 (0–0)	<0.001
HGS (kg)	15.8 (7.4–25.6)	22.7 (16.5–31.5)	6.4 (0–11.0)	<0.001
Laboratory data				
SPD (µM)	5.93 (4.60–7.68)	5.63 (4.36–7.05)	7.02(5.18–8.69)	0.002
SPM (µM)	3.15 (2.67–3.98)	3.19 (2.62–4.07)	3.04 (2.72–3.91)	0.701
SPM/SPD	0.54 (0.43–0.70)	0.57 (0.46–0.75)	0.49 (0.39–0.60)	<0.001
Hb (g/dL)	12.9 (11.6–13.9)	13.5 (12.0–14.3)	12.0 (11.3–13.2)	<0.001
Alb (g/dL)	4.0 (3.6–4.3)	4.2 (4.0–4.4) ^#^	3.5 (3.3–3.8)	<0.001
eGFR (mL/min/1.73 m^2^)	58.0 (47.5–69.7)	57.0 (45.9–66.8)	60.0 (48.1–80.2)	0.046
LDL-C (mg/dL)	111 (95–129)	111 (94–128)	106 (97–131)	0.661
HDL-C (mg/dL)	60 (47–72)	61 (47–74)	57 (46–69)	0.179
TG (mg/dL)	113 (84–160)	128 (90–184)	100 (76–131)	<0.001
HbA1c (%)	5.8 (5.6–6.2)	6.0 (5.7–6.4) ^#^	5.6 (5.4–5.8)	<0.001

Values are expressed as median (interquartile range). BMI data are expressed as mean ± SD. * The Mann–Whitney *U* test was performed between the non-sarcopenia and sarcopenia groups in each variable, and the results are shown with *p* values. As BMI was normally distributed, the unpaired Student’s *t*-test was used to compare the two groups. No. of N.H. residents—number of nursing home residents; CVDs—cardiovascular diseases; CeVDs—cerebrovascular diseases; BMI—body mass index; SD—standard deviation; SMI—skeletal muscle mass index; WS—walking speed; HGS—hand grip strength; SPD—spermidine; SPM—spermine; Hb—hemoglobin; Alb—albumin; eGFR—estimated glomerular filtration rate; LDL-C—low-density lipoprotein cholesterol; HDL-C—high-density lipoprotein cholesterol; TG—triglyceride. ^#^
*n* = 110 due to missing measurements.

**Table 2 biomedicines-11-01403-t002:** Binomial multivariate logistic regression analysis including SPD and SPM concentrations.

Variables	OR	OR (95% CI)	*p* Value
Sex (M:1/F:0)	1.677	0.455–6.181	0.437
Age	1.115	1.021–217	0.016
CVDs	0.631	0.028–14.126	0.772
CeVDs	0.910	0.188–4.415	0.907
Dementia	5.427	1.089–27.044	0.039
Bone fracture	0.321	0.054–1.900	0.210
BMI	0.730	0.573–0.930	0.011
SPD	1.481	1.073–2.044	0.017
SPM	0.502	0.299–0.842	0.009
Hb	0.842	0.598–1.185	0.323
Alb	0.115	0.019–0.704	0.019
eGFR	1.050	1.011–1.089	0.010
LDL-C	0.984	0.959–1.010	0.219
HDL-C	0.994	0.953–1.036	0.768
TG	0.998	0.989–1.007	0.690
HbA1c	0.241	0.078–0.744	0.013

The non-sarcopenia group was used as the reference for the analysis. The units for each value are given in the text. The diagnostic criteria for sarcopenia are described in the main text. CVDs—cardiovascular diseases; CeVDs—cerebrovascular diseases; BMI—body mass index; SPD—spermidine; SPM—spermine; Hb—hemoglobin; Alb—albumin; eGFR—estimated glomerular filtration rate; LDL-C—low-density lipoprotein cholesterol; HDL-C—high-density lipoprotein cholesterol; TG—triglyceride; OR—odds ratio; 95% CI—95% confidence interval.

**Table 3 biomedicines-11-01403-t003:** Binomial multivariate logistic regression analysis including the SPM/SPD ratio.

Variables	OR	95% CI	*p* Value
Sex (M:1/F:0)	1.759	0.487–6.355	0.389
Age	1.114	1.024–1.211	0.012
CVDs	0.615	0.030–12.811	0.754
CeVDs	1.059	0.230–4.870	0.941
Dementia	4.213	0.998–17.789	0.050
Bone fracture	0.356	0.064–1.969	0.237
BMI	0.720	0.566–0.917	0.008
SPM/SPD	0.033	0.002–0.557	0.018
Hb	0.829	0.574–1.198	0.318
Alb	0.120	0.020–0.710	0.019
eGFR	1.048	1.010–1.086	0.011
LDL-C	0.985	0.961–1.009	0.221
HDL-C	0.996	0.956–1.037	0.830
TG	0.998	0.990–1.007	0.693
HbA1c	0.240	0.083–0.697	0.009

The non-sarcopenia group was used as the reference for analysis. The units for each value are given in the text. The diagnostic criteria for sarcopenia are described in the main text. CVDs—cardiovascular diseases; CeVDs—cerebrovascular diseases; BMI—body mass index; SPD—spermidine; SPM—spermine; Hb—hemoglobin; Alb—albumin; eGFR—estimated glomerular filtration rate; LDL-C—low-density lipoprotein cholesterol; HDL-C—high-density lipoprotein cholesterol.

**Table 4 biomedicines-11-01403-t004:** Characteristics of the healthy, semi-sarcopenia, and sarcopenia groups.

Characteristics	Healthy	Semi-Sarcopenia		Sarcopenia		
			*p* Value ^†^		*p* Value ^†^	*p* Value ^††^
No. of patients (M/F)	56 (37/19)	55 (15/40)	<0.05	71 (16/55)	<0.05	n.s
Age (years)	73 (71–79)	84 (77–88)	<0.001	90 (84–95)	<0.001	<0.001
No. of N.H. residents (%)	0 (0)	6 (11)	<0.05	57 (80)	<0.05	<0.05
Past history (%)						
CVDs	2 (4)	4 (7)	n.s	2 (3)	n.s	n.s
CeVDs	2 (4)	10 (18)	<0.05	27 (38)	<0.05	<0.05
Dementia, n (%)	1 (2)	15 (27)	<0.05	55 (77)	<0.05	<0.05
Bone fracture	2 (4)	6 (11)	n.s	15 (21)	<0.05	n.s
BMI (kg/m^2^)	23.4 ± 2.5	23.7 ± 3.0	0.915	19.7 ± 3.1	<0.001	<0.001
SMI (kg/m^2^)	8.3 (7.6–8.9)	6.9 (6.2–8.0)	<0.001	4.6 (3.6–5.3)	<0.001	<0.001
WS (m/s)	1.25 (1.14–1.35)	0.87 (0.69–0.98)	<0.001	0 (0–0)	<0.001	<0.001
HGS (kg)	30.0 (23.2–34.2)	16.7 (12.1–20.3)	<0.001	6.4 (0–11.0)	<0.001	<0.001
Laboratory findings						
SPD (µM)	5.07 (3.96–6.74)	5.95 (4.97–7.74)	0.087	7.02 (5.18–8.69)	<0.001	0.087
SPM (µM)	3.05 (2.43–3.85)	3.29 (2.92–4.49)	0.150	3.04 (2.72–3.91)	0.894	0.281
SPM/SPD	0.60 (0.49–0.75)	0.55 (0.43–0.75)	0.722	0.49 (0.39–0.60)	0.010	0.087
Hb (g/dL)	13.9 (13.0–14.6)	12.7 (11.5–13.8)	0.008	12.0 (11.3–13.2)	<0.001	0.079
Alb (g/dL)	4.2 (4.1–4.4) ^a^	4.1 (3.8–4.4)	0.010	3.5 (3.3–3.8)	<0.001	<0.001
eGFR (mL/min/1.73 m^2^)	60.5 (49.7–70.9)	54.4 (42.4–60.2)	0.023	60.0 (48.1–80.2)	0.556	0.003
LDL-C (mg/dL)	111 (94–129)	111 (94–128)	0.988	106 (97–131)	0.616	0.715
HDL-C (mg/dL)	63 (50–76)	60 (46–74)	0.619	57 (46–69)	0.205	0.756
TG (mg/dL)	128 (85–193)	128 (93–179)	0.803	100 (76–131)	0.046	0.035
HbA1c (%)	6.1 (5.7–6.6)	6.0 (5.6–6.3) ^b^	0.207	5.6 (5.4–5.8)	<0.001	0.002

Values are expressed as median (interquartile range). BMI data are expressed as mean ± SD. The z-test was used for multiple comparisons of proportions, and the Bonferroni method was used to determine significance. Specifically, the Bonferroni method was used to obtain the “adjusted significance level (p’)”, and a judgment was made at p’ = 0.017 for the probability value of the significance test (z-test) results for each comparison pair. The results of group comparisons judged to be significantly different at p’ were noted as *p* < 0.05 and are listed. The Kruskal–Wallis test was used for multiple comparisons of values. Levene’s test was used to evaluate the homogeneity of variance. Tukey’s method was used to determine between-group differences for variables with equal variance, and Games-Howell’s method was used for variables without equal variance. ^†^ Comparison results with the healthy group. ^††^ Comparison results with the semi-sarcopenia group No. of N.H. residents—number of nursing home residents; CVDs—cardiovascular diseases; CeVDs—cerebrovascular diseases; BMI—body mass index; SD—standard deviation; SMI—skeletal muscle mass index; WS—walking speed; HGS—hand grip strength; SPD—spermidine; SPM—spermine; Hb—hemoglobin; Alb—albumin; eGFR—estimated glomerular filtration rate; LDL-C—low-density lipoprotein cholesterol; HDL-C—high-density lipoprotein cholesterol; TG—triglyceride; n.s.—not significance. Criteria for the classification of healthy, semi-sarcopenia, and sarcopenia groups are listed in the text. ^a^
*n* = 55 and ^b^
*n* = 54 due to missing measurements.

**Table 5 biomedicines-11-01403-t005:** Multinomial multivariate logistic regression analysis including SPD and SPM concentrations.

Variables	Semi-Sarcopenia (*n* = 54) ^a^			Sarcopenia (*n* = 71)		
	OR	95% CI	*p*-Value	OR	95% CI	*p*-Value
Sex (M:1/F:0)	0.090	0.021–0.378	0.001	0.259	0.044–1.533	0.136
Age	1.329	1.159–1.524	<0.001	1.403	1.203–1.636	<0.001
CVDs	6.635	0.274–160.6	0.245	4.637	0.068–315.6	0.476
CeVDs	18.94	1.813–197.9	0.014	12.63	0.855–186.6	0.065
Dementia	3.092	0.224–42.66	0.399	14.80	0.792–276.8	0.071
Bone fracture	0.696	0.039–12.29	0.805	0.188	0.007–4.988	0.317
BMI	1.068	0.795–1.435	0.662	0.750	0.523–1.076	0.118
SPD	1.625	1.110–2.379	0.013	2.218	1.393–3.533	0.001
SPM	1.024	0.528–1.986	0.944	0.487	0.214–1.105	0.085
Hb	1.033	0.732–1.456	0.854	0.811	0.540–1.217	0.311
Alb	0.411	0.042–4.068	0.447	0.040	0.003–0.617	0.021
eGFR	1.004	0.959–1.052	0.852	1.066	1.006–1.129	0.030
LDL-C	0.981	0.954–1.008	0.171	0.964	0.930–0.999	0.042
HDL-C	1.006	0.963–1.051	0.794	1.003	0.948–1.060	0.930
TG	1.008	1.000–1.016	0.065	1.006	0.995–1.018	0.265
HbA1c	0.389	0.137–1.110	0.078	0.087	0.019–0.389	0.001

The healthy group (participants without sarcopenia or semi-sarcopenia) was used as the reference for the analysis. The units for each value are given in the text. The diagnostic criteria for non-sarcopenia, semi-sarcopenia, and sarcopenia are described in the main text. CVDs—cardiovascular diseases; CeVDs—cerebrovascular diseases; BMI—body mass index; SPD—spermidine; SPM—spermine; Hb—hemoglobin; Alb—albumin; eGFR—estimated glomerular filtration rate; LDL-C—low-density lipoprotein cholesterol; HDL-C—high-density lipoprotein cholesterol; TG—triglyceride; OR—odds ratio; 95% CI—95% confidence interval. ^a^
*n* = 54 due to a missing measurement of HbA1c.

**Table 6 biomedicines-11-01403-t006:** Multinomial multivariate logistic regression analysis including the SPM/SPD ratio.

Variables	Semi-Sarcopenia (*n* = 54) ^a^			Sarcopenia (*n* = 71)		
	OR	95% CI	*p*-Value	OR	95% CI	*p*-Value
Sex (M:1/F:0)	0.104	0.026–0.422	0.002	0.298	0.052–1.703	0.173
Age	1.281	1.132–1.450	<0.001	1.358	1.178–1.565	<0.001
CVDs	7.036	0.267–185.4	0.242	5.703	0.085–384.3	0.418
CeVDs	26.54	2.065–341.1	0.012	21.12	1.176–379.3	0.038
Dementia	7.408	0.491–111.8	0.148	24.28	1.328–443.7	0.031
Bone fracture	0.953	0.053–17.04	0.974	0.290	0.012–7.302	0.452
BMI	1.081	0.804–1.453	0.608	0.753	0.522–1.085	0.128
SPM/SPD	0.057	0.004–0.796	0.033	0.002	<0.001–0.091	0.001
Hb	1.017	0.688–1.502	0.934	0.779	0.488–1.244	0.295
Alb	0.392	0.042–3.695	0.413	0.043	0.003–0.628	0.021
eGFR	0.999	0.957–1.043	0.958	1.054	0.999–1.111	0.055
LDL-C	0.981	0.955–1.007	0.156	0.967	0.934–1.000	0.051
HDL-C	1.006	0.965–1.050	0.769	1.006	0.953–1.062	0.822
TG	1.010	1.002–1.018	0.016	1.008	0.998–1.019	0.125
HbA1c	0.486	0.177–1.337	0.162	0.110	0.026–0.459	0.002

The healthy group (participants without sarcopenia or semi-sarcopenia) was used as the reference for the analysis. The units for each value are given in the text. The diagnostic criteria for non-sarcopenia, semi-sarcopenia, and sarcopenia are described in the main text. CVDs—cardiovascular diseases; CeVDs—cerebrovascular diseases; BMI—body mass index; SPD—spermidine; SPM—spermine; Hb—hemoglobin; Alb—albumin; eGFR—estimated glomerular filtration rate; LDL-C—low-density lipoprotein cholesterol; HDL-C—high-density lipoprotein cholesterol; TG—triglyceride; OR—odds ratio; 95% CI—95% confidence interval ^a^
*n* = 54 due to a missing measurement of HbA1c.

## Data Availability

The data that support the findings of this study are available from the corresponding author (Kuniyasu Soda) upon reasonable request.

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
