# Peer review of "Whole Blood Spermine/Spermidine Ratio as a New Indicator of Sarcopenia Status in Older Adults"

_biomedicines, 2023, doi:10.3390/biomedicines11051403_

Round 1
Reviewer 1 Report
The manuscript of Sanayama et al. (Biomedicines-2361606) is of high priority and points to the problems and enigmas of (i) polyamine (PA) concentration in biological tissue, (ii) ratio of key ultimate PAs, spermine/spermidine, and (iii) their relationship to a specific disorder related to aging. These three points are the novelty of the current manuscript since there is no clarity on it in the literature. Testing spermine/spermidine ratio (SSR) authors showed that the SSR in blood samples may be a diagnostic parameter for indication of muscle mass loss and weakness (sacropenia) and aging. Methods and statistics are appropriate. I recommend that the authors could write a brief paragraph such as Conclusions because it is not enough stated in the Abstract.
Author Response
Thank you very much for your kind suggestion. We have added a conclusion section at the end of the manuscript. Please evaluate it.
- Conclusions
The study first showed the inverse relationship between blood spermine concentration relative to spermidine concentration (SPM/SPD ratio) and the degree of sarcopenia. And we also discussed the possible pathological background. Since several reports showed the inverse relationship between blood SPM/SPD ratio and the degree of age-related diseases, blood SPM/SPD ratio may be an indicator to predict or detect age-related pathological changes.
Reviewer 2 Report
This is a well-written manuscript, but I have a number of suggestions for revision.
Abstract, line 27: If the word count permits, please briefly indicate the measures used to define sarcopenia from the Asian Working Group for Sarcopenia.
Line 29: Please define the abbreviations SPM/SPD the first time you use them.
Line 39: Here in the opening of your paper, you mention that sarcopenia is a risk factor for a number of conditions, but you haven’t defined sarcopenia. I suggest you provide a definition in the first sentence of your introduction.
Line 69: Please give examples of some food sources that are high in polyamines
In the introduction you haven’t really mentioned why polyamine levels might affect muscle mass. How could these levels theoretically affect muscle mass? For example, you mention effects on inflammation, which might in turn affect muscle protein degradation and protein synthesis. Please include this link in your introduction.
The end of the introduction needs a hypothesis statement.
Line 184: Please provide the %CVs (coefficients of variation) for these assays.
Line 231: “CeVDs” – please make sure this abbreviation is defined
Line 233: “BMI, SMI, WS, and HGS, which are the diagnostic elements of sarcopenia...” – I don’t think BMI is a diagnostic element.
Figure 3: I think this figure can be deleted since all correlations are non-significant. I think it is sufficient to just present the correlations in the text of the results.
Line 292: “To estimate the risk of developing sarcopenia, a binomial multivariate logistic regression analysis was performed on each value for the non-sarcopenia group. Because SMI, HGS, and WS are diagnostic criteria for sarcopenia and nursing home occupancy rates reflect the consequences of developing of sarcopenia, analyses were performed after excluding these factors.” – two comments here: 1) this should be moved to the statistics section; 2) It is unclear what you are doing here...please consider re-wording.
Tables 2 and 3: Here is it clear that the non-sarcopenic group is used as a reference...perhaps clarify this in the above description. The footnote to this table mentions sarcopenic group and semi-sarcopenic group. Are the sarcopenic and semi-sarcopenic groups combined here or is the semi-sarcopenic group part of the non-sarcopenic reference group?
Line 511: it is stated it is difficult to identify a cut-off level at which blood polyamine levels would be predictive of sarcopenia. Is this also true of the ratio (SPM/SPD)? Do you think a specific ratio could be used as a cut-off to identify sarcopenia risk?
Author Response
Thank you very much for your kind suggestion. Corrected text is shown in red.
Comments: Abstract, line 27: If the word count permits, please briefly indicate the measures used to define sarcopenia from the Asian Working Group for Sarcopenia.
Responses: We replaced the sentence with “Sarcopenia was determined by muscle mass, muscle strength, and physical performance based on the criteria of the Asian Working Group for Sarcopenia (2019).” And the word count in the abstract is 200.
Comments: Line 29: Please define the abbreviations SPM/SPD the first time you use them.
Responses: SPM/SPD in the abstract was replaced by spermine/spermidine.
And, we have replaced the words "SPM/SPD ratio" in line 65 with "Blood SPM concentration relative to blood SPD (SPM/SPD ratio)". (line 68 to 69 in the revised manuscript.)
Comments: Line 39: Here in the opening of your paper, you mention that sarcopenia is a risk factor for a number of conditions, but you haven’t defined sarcopenia. I suggest you provide a definition in the first sentence of your introduction.
Responses: We have added the definition of sarcopenia in the first sentence of the Introduction. The sentence has been replaced with “Sarcopenia, defined as the age-related loss of muscle mass, strength, and physical performance, is a progressive, generalized musculoskeletal disorder that is associated with an increased risk of adverse events, such as falls, fractures, and impaired mobility [1].
Comments: Line 69: Please give examples of some food sources that are high in polyamines
Responses: We used Natto (fermented soybeans) in our previous human intervention study, and chow supplemented with synthetic polyamine in the mouse study. Therefore, we do not feel it is appropriate to list examples of polyamine-rich foods here. Because we have already published original articles and reviews defining high-polyamine diets, authors can easily find polyamine-rich foods.
Comments: (Line 93 – 99): In the introduction you haven’t really mentioned why polyamine levels might affect muscle mass. How could these levels theoretically affect muscle mass? For example, you mention effects on inflammation, which might in turn affect muscle protein degradation and protein synthesis. Please include this link in your introduction. The end of the introduction needs a hypothesis statement.
Responses: We have changed the last paragraph of the Introduction to read as follows. “Sarcopenia is a typical condition that develops and progresses with age. While previous research has shown that blood polyamines are associated with age-related diseases, no previous study has examined whether they are associated with the typical age-related pathological condition, sarcopenia. In addition, a simple diagnostic approach to predict the risk of developing sarcopenia has not yet been developed. Therefore, in this study, we investigated whether blood polyamine levels are associated with sarcopenia and whether blood polyamine levels can be used as a biomarker for the early detection or assessment of progression of sarcopenia.”
Comments: Line 184: Please provide the %CVs (coefficients of variation) for these assays.
Responses: The description about the %CVs (coefficients of variation) in this assay was added calculated and specified in the manuscript. In line 184; Blood SPD, SPM, and SPM/SPD ratio were each measured three consecutive times in two samples and the average was calculated as the %CV (coefficient of variation). The %CVs were 4.3% for SPD, 3.6% for SPM, and 6.7% for the SPM/SPD ratio. (from line 189 to 191 in the revised manuscript.)
Comments: Line 231: “CeVDs” – please make sure this abbreviation is defined
Responses: CeVD is defined in line 113 of the original manuscript. CeVDs is the plural form of CeDV, just as we used CVDs as the plural of CVD.
Comments: Line 233: “BMI, SMI, WS, and HGS, which are the diagnostic elements of sarcopenia...” – I don’t think BMI is a diagnostic element.
Responses: Thank you for your point. Your point is correct. We replaced the sentence with "SMI, WS, and HGS, which are the diagnostic elements of sarcopenia and its associated indices, as well as BMI, were all significantly lower (p < 0.001) in the sarcopenia group than in the non-sarcopenia group.” (from line 243 to 245 of the revised manuscript.)
Comments: Figure 3: I think this figure can be deleted since all correlations are non-significant. I think it is sufficient to just present the correlations in the text of the results.
Responses: While certainly not significant, this number is important and we need to make the other researchers aware of it. The reason for this is to visually clarify the difference between the change in polyamine concentration in healthy people as they age and the change in polyamine concentration in people with sarcopenia as they age. We believe it is very important for polyamine researchers to recognize these figures.
Comments: Line 292: “To estimate the risk of developing sarcopenia, a binomial multivariate logistic regression analysis was performed on each value for the non-sarcopenia group. Because SMI, HGS, and WS are diagnostic criteria for sarcopenia and nursing home occupancy rates reflect the consequences of developing of sarcopenia, analyses were performed after excluding these factors.” – two comments here: 1) this should be moved to the statistics section; 2) It is unclear what you are doing here...please consider re-wording.
Responses:
- We moved the sentences you pointed out “Because SMI, HGS, and WS are diagnostic criteria for sarcopenia and nursing home occupancy rates reflect the consequences of developing of sarcopenia, analyses were performed after excluding these factors.” to the statistics section (from line 214 to line 216 of the revised manuscript)
- Binomial Multivariate Logistic Regression Analysis: We have replaced the begging in this section with the following sentences. “Binomial multivariate logistic regression analysis was performed to select parameters associated with sarcopenia progression. The non-sarcopenia group was used as the reference for the analysis, and ORs (95% confidence intervals) were calculated for each explanatory variable. ORs were calculated using all explanatory variables listed in Tables 2 and 3. (from line 302 to 306 in the revised manuscript.)
Comments: Tables 2 and 3: Here is it clear that the non-sarcopenic group is used as a reference...perhaps clarify this in the above description. The footnote to this table mentions sarcopenic group and semi-sarcopenic group. Are the sarcopenic and semi-sarcopenic groups combined here or is the semi-sarcopenic group part of the non-sarcopenic reference group?
Responses: Thank you very much for your pointing out. The description of the footnotes to Table 2 and Table 3 to which you referred, "The diagnostic criteria for non-sarcopenia, semi-sarcopenia, and sarcopenia are described in the main text" has been replaced with "The diagnostic criteria for sarcopenia are described in the main text.
Tables 2 and 3: Here is it clear that the non-sarcopenic group is used as a reference...perhaps clarify this in the above description. The footnote to this table mentions sarcopenic group and semi-sarcopenic group. Are the sarcopenic and semi-sarcopenic groups combined here or is the semi-sarcopenic group part of the non-sarcopenic reference group?
Responses: The description in the footnote of Tabel 2 and 3 were error due to our mistake. The analyses were performed with non-sarcopenia (including healthy and semi-sarcopenia groups) and sarcopenia. The descriptions were replaced by “The diagnostic criteria for sarcopenia are described in the main text.”
Comments: Line 511: it is stated it is difficult to identify a cut-off level at which blood polyamine levels would be predictive of sarcopenia. Is this also true of the ratio (SPM/SPD)? Do you think a specific ratio could be used as a cut-off to identify sarcopenia risk?
Responses: SPM/SPD also has some degree of variability, making it difficult to set a cutoff value. While the search for cutoff values is important and should be addressed, at this time we believe it would be useful to examine their change over time.